# Promising Effects of Digital Chest Tube Drainage System for Pulmonary Resection: A Systematic Review and Network Meta-Analysis

**DOI:** 10.3390/jpm12040512

**Published:** 2022-03-22

**Authors:** Po-Chih Chang, Kai-Hua Chen, Hong-Jie Jhou, Cho-Hao Lee, Shah-Hwa Chou, Po-Huang Chen, Ting-Wei Chang

**Affiliations:** 1Division of Thoracic Surgery, Department of Surgery, Kaohsiung Medical University Hospital, Kaohsiung Medical University, Kaohsiung City 80708, Taiwan; dyno910076@hotmail.com (P.-C.C.); shhwch@kmu.edu.tw (S.-H.C.); 2Weight Management Center, Kaohsiung Medical University Hospital, Kaohsiung Medical University, Kaohsiung City 80708, Taiwan; 3Ph.D. Program in Biomedical Engineering, College of Medicine, Kaohsiung Medical University, Kaohsiung City 80708, Taiwan; 4Department of Sports Medicine, College of Medicine, Kaohsiung Medical University, Kaohsiung City 80708, Taiwan; 5Department of Surgery, Kaohsiung Medical University Hospital, Kaohsiung Medical University, Kaohsiung City 80708, Taiwan; dr.chenkaihua@outlook.com; 6Department of Neurology, Changhua Christian Hospital, Changhua 50006, Taiwan; xsai4295@gmail.com; 7Division of Hematology and Oncology Medicine, Department of Internal Medicine, National Defense Medical Center, Tri-Service General Hospital, Taipei City 11490, Taiwan; drleechohao@gmail.com; 8Department of Surgery, School of Medicine, College of Medicine, Kaohsiung Medical University, Kaohsiung City 80708, Taiwan; 9Department of Internal Medicine, Tri-Service General Hospital, National Defense Medical Center, Taipei City 11490, Taiwan

**Keywords:** digital chest tube, chest tube drainage system, lung resection, pulmonary resection, meta-analysis, network meta-analysis

## Abstract

Objective: The chest tube drainage system (CTDS) of choice for the pleural cavity after pulmonary resection remains controversial. This systematic review and network meta-analysis (NMA) aimed to assess the length of hospital stay, chest tube placement duration, and prolonged air leak among different types of CTDS. Methods: This systemic review and NMA included 21 randomized controlled trials (3399 patients) in PubMed and Embase until 1 June 2021. We performed a frequentist random effect in our NMA, and a P-score was adopted to determine the best treatment. We assessed the clinical efficacy of different CTDSs (digital/suction/non-suction) using the length of hospital stay, chest tube placement duration, and presence of prolonged air leak. Results: Based on the NMA, digital CTDS was the most beneficial intervention for the length of hospital stay, being 1.4 days less than that of suction CTDS (mean difference (MD): −1.40; 95% confidence interval (CI): −2.20 to −0.60). Digital CTDS also had significantly reduced chest tube placement duration, being 0.68 days less than that of suction CTDSs (MD: −0.68; 95% CI: −1.32 to −0.04). Neither digital nor non-suction CTDS significantly reduced the risk of prolonged air leak. Conclusions: Digital CTDS is associated with better outcomes than suction and non-suction CTDS for patients undergoing pulmonary resections, specifically 0.68 days shorter chest tube duration and 1.4 days shorter hospital stay than suction CTDS.

## 1. Introduction

Pulmonary resection remains the mainstay diagnostic or therapeutic solution for associated pathologic entities. An adequate drainage of the pleural space is essential for a successful pulmonary resection. Since 1922, placing a chest tube for drainage has been the gold standard management for pulmonary resections [1]. Aside from draining intrapleural fluid and air, an ideal chest tube drainage system (CTDS) can help clinicians detect persistent air leak and evacuate the pneumothorax or hemothorax concomitantly [2]. Based on the idea of an enhanced recovery after surgery, an earlier removal can eventually minimize the associated pain, shorten the hospital stay, and reduce hospital costs [3].

In the current era, many CTDSs have been clinically adopted, including a digital suction system, analog/traditional suction system, and water seal. A water seal (no suction) CTDS is the original, less costly modality for those with no to minimal air leak after pulmonary resections [2]. To facilitate the pulmonary expansion and eliminate the residual space, applying a suction CTDS (water seal with external suction) could be an option to evacuate the air via negative pressure for those with prolonged air leaks after pulmonary resections or for those prone to have air leaks through the suture/staple lines due to emphysema, major anatomic pulmonary resection, or pleural adhesions [2,3,4]. Since 2007, some postulated that a digital CTDS could reduce the inter-observer variability in air leak, fluid, and intrapleural pressure assessment and maintain the predetermined negative intrapleural pressure via an electronic sensor and digital console [2,3,5]. Due to its precise measurement with steady intrapleural environment maintenance within 0.1 cm H_2_O, the digital chest tube drainage system (CTDS) may be associated with quicker chest tube removal, shorter hospital stay, and higher satisfaction than those managed with water seal CTDS among patients undergoing pulmonary resection [6,7,8,9,10].

Given the fact that thoracic surgeons have individualized experiences and preferences for choosing different devices, the proper CTDS after pulmonary resection still remains controversial [7,8,9,10,11,12,13,14,15,16,17,18,19,20,21,22,23,24,25,26,27]. Previous meta-analysis studies have only reported pairwise comparisons [28,29,30,31,32]. Thus, it is necessary to collate the results of randomized controlled trials to compare three interventions simultaneously in a single analysis and to estimate the ranking and hierarchy of interventions. Herein, a network meta-analysis was conducted to catalog the results of these RCTs into a comprehensive systematic review and meta-analysis to determine the proper CTDS based on the length of hospital stay, chest tube placement duration, presence of prolonged air leak, and associated adverse events.

## 2. Methods

This systematic review with network meta-analysis was conducted according to the Preferred Reporting Items for Systematic Reviews and Meta-Analyses (PRISMA) for Network Meta-Analyses [33] (Appendix A). The protocol for the systemic review was pre-specified and we registered the protocol in the Open Science Framework (protocol available at https://osf.io/ep3b9 accessed on 13 July 2021).

### 2.1. Search Strategy

A comprehensive search was performed without language restrictions using Cochrane Database, Embase and PubMed from inception until 1 June 2021 to search for all unpublished and published trials, with reviewing of abstracts and screening of titles. The search terms used for above-mentioned databases were keywords involving “pulmonary resection”, “lung surgery”, and combinations of “chest tube drainage system”, “devices”, or “suction” (Appendix A). A manual search of gray literature (doctoral theses and conference abstracts) and reference lists of included articles was conducted to ensure that no studies were missed.

### 2.2. Study Selection

We included parallel-group randomized studies on patients undergoing pulmonary resection with various demographics (aged ≥ 18 years; with pulmonary diseases such as primary lung cancer, metastatic lung cancer, benign lung tumors, or other pulmonary diseases requiring resection; receiving different pulmonary resections, including bilobectomy, lobectomy, segmentectomy, or wedge resection), reporting at least one interested outcome and undergoing non-suction, suction, or digital CTDSs. The exclusion criteria include those with a single-arm study only or without randomized-control design, patients’ age < 18 years, or incompletely reported essential outcomes or information. Those with chest tube placement without pulmonary resection were also excluded.

Three different types of CTDS were identified and analyzed: digital, suction, and non-suction CTDS. Digital CTDS had digital sensors continuously monitoring pleural pressure and air flow, and applied an external suction accordingly, which would keep pleural pressure in a steady state. Suction CTDS had a fixed external suction force. Non-suction CTDS included those without an external suction force. The type of CTDS used in each study is described in Appendix A.

Two review authors independently reviewed the trials based on the eligibility criteria by screening their abstracts and titles, and another author adjudicated the differences. A disagreement in the study selection was resolved by group discussion to have a final decision.

### 2.3. Data Extraction and Bias Assessment

Two reviewers independently assessed the eligibility of all extracted data and identified citations. Then, the two reviewers performed data extraction with a specifically designed form to capture study-related (study design, author name, nation and publication year), participant-related (participant’s characteristics, sample size, and measurement tools), and intervention-related characteristics (Appendix A). An intention-to-treat principle was followed to extract the participant-related information in the enrolled RCTs. For relevant data that were unclearly reported or missing, the corresponding authors were conducted for the required information.

The methodological quality of each study was assessed independently by the two reviewers using the Cochrane Collaboration tool for RCTs [34]. If discrepancies in the quality appraisal or data extraction existed, a consensus was achieved by consulting a third reviewer or by deliberating within the group.

### 2.4. Outcome Measures

The data were extracted to consolidate them into one primary outcome and two secondary outcomes. The primary outcome was the length of hospital stay, defined as the days that the enrolled patients stay in a hospital after pulmonary resections. The secondary outcomes were as follows: chest tube placement duration (defined as the interval between chest tube insertion after pulmonary resections and removal), presence of prolonged air leak (defined as air leak noted longer than three postoperative days based on the enrolled studies) [7,8,10,13,14,15,16,17,19,20,21,24,27,35] (Table 1).

### 2.5. Data Synthesis and Statistical Analysis

This network meta-analysis was conducted using a frequentist approach [36,37]. This provided a point estimate from the network and a 95% confidence interval (CI) from the frequency distribution of the estimate. All network meta-analyses were conducted using the statistical package “netmeta” 0.9–0 (https://cran.rproject.org/web/packages/netmeta/i-ndex.html, accessed on 1 June 2021.) in R 4.1.0 (R Core Team, Vienna, Austria) and Stata version 16 (Stata Corp, College Station, Texas) [38].

The symmetry and geometry of the evidence were examined by performing a network plot with nodes for the study subjects. The contribution of each direct comparison to the estimation of network summary effects was calculated based on the combination of variances in direct treatment effects and network structure [39]. The comparisons between the direct and indirect evidence in network meta-analysis were also summarized (Appendix A). The pooled odds ratio (OR) with 95% CIs and mean differences (MDs) with 95% Cis [39] were calculated for dichotomous and continuous variables, respectively, to summarize the effects of each comparison using a random-effects model for variations across studies.

Digital and non-suction CTDS were ranked by their probability of being the best treatment compared with suction CTDS. In the frequentist network meta-analysis, a P-score was adopted, which was based solely on the point estimates and standard errors of the network estimates; it also demonstrated the percentage of being the best treatment [40,41,42]. A higher P-score indicated a shorter hospital stay length, shorter chest tube placement duration, and lower risk of prolonged air leak [43]. Forest plots represented the summary of these results within the network meta-analysis, including the relative mean effects, 95% CIs, and P-scores for all interventions [44]. The values of P-score were shown in a rank-heat plot, which disclosed the probability of being the best intervention in relation to all outcomes [45].

Potential inconsistencies could be observed when direct and indirect effects are incongruent within the same comparison in the network. A random-effects design-by-treatment interaction model and a node-splitting technique were used for each comparison to identify the inconsistencies [46,47,48,49]. Statistical significance was defined as *p* < 0.05 for both analyses. Sensitivity analyses were performed to examine the validity of study findings by omitting high risk of bias studies.

Among the articles in our network meta-analysis (NMA), there should be an assumption of transitivity in terms of clinical characteristics and the methodology employed, such as regarding the differences in the characteristics of enrolled patients, study designs, interventions, and outcome measurements. Transitivity cannot be statistically tested, but this should be conceptually considered by reviewing the distributions of potential confounding factors across studies.

Comparison-adjusted funnel plots were created, and Egger’s test was applied to evaluate small study effects and publication biases [40,50].

### 2.6. Quality Assessment

The Grading of Recommendations, Assessment, Development, and Evaluation (GRADE) criteria were used for the quality of evidence in this network meta-analysis [49,51]. First, the quality of indirect and direct evidence was assessed separately. If evidence was derived from the RCTs, a direct evidence rating was initially high, but it might be downrated by any concerns regarding the risk of bias (RoB), indirectness, inconsistency, publication bias or imprecision. An indirect evidence rating started from a lower rating in two pairwise direct comparisons of first-order loops, but it could be downrated based on intransitivity (differences between studies that represented the basis of methodological characteristics or indirect evidence for clinical) or imprecision. Then, when both direct and indirect evidence were available, the quality rating that was higher was recommend by the GRADE Working Group as the preferable rating of the quality of effect estimates in this network meta-analysis [49,51] (Appendix A).

## 3. Results

### 3.1. Systematic Literature Review

The beginning screening obtained 1913 article titles and abstracts via the electronic databases. In total, 1278 citations remained after removing duplicates. From those, 1241 studies were excluded by further screening using the title or abstract to meet the clinical trial requirement. Then, 37 full texts were assessed for potential eligibility, and 16 studies were excluded for various reasons, such as limited data without available results, enrolled patients without pulmonary resections or operation, no comparison CTDS, or retrospective cohort study only. The remaining 21 studies were randomized controlled trials and reported as complete research articles, and our quantitative synthesis included these studies [7,8,9,10,11,12,13,14,15,16,17,18,19,20,21,22,23,24,25,26,27] (Figure 1).

These 21 RCTs investigated a total of 3399 participants randomized to the following interventions after thoracic surgery: non-suction, suction, or digital CTDSs [7,8,9,10,11,12,13,14,15,16,17,18,19,20,21,22,23,24,25,26,27].

The study characteristics are summarized in Table 1 and Appendix A. The sample sizes of the studies ranged from 31 to 500 patients, of which 58.9% were males. The mean age of the subjects was 63.2 years. In the reported surgical approaches (*n* = 2326), 887 patients underwent video-assisted thoracoscopic surgery (VATS) (38.13%), and 1439 patients underwent thoracotomy (61.87%). As for the resection type (*n* = 2744), 2089 patients underwent lobectomy or bilobectomy (76.13%), 189 patients underwent segmentectomy (6.89%), and 466 patients underwent wedge resection or lung biopsy (16.98%).

### 3.2. Results of Hospital Stay Length

In terms of the primary outcome, which was the length of hospital stay, 15 studies were included with particular consideration of transitivity conception and a total of 1870 participants and three intervention options [8,9,11,12,13,15,17,18,19,21,22,23,25,26,27] (Figure 2A). All interventions were sorted based on their ranking and were accompanied by MDs and 95% CIs versus the comparator “suction CTDS.” Digital and non-suction CTDSs were significantly in association with shorter hospital stay than suction CTDS, with MD ranging between −1.40 (95% CI: −2.20–−0.60) for digital CTDS and −1.05 (95% CI: −1.91–−0.18) for non-suction CTDS (Figure 3A). The P-score of digital CTDS at 0.90 was the highest (Figure 4). Appendix A shows the details of the head-to-head comparisons of outcomes.

### 3.3. Results of Chest Tube Placement Duration

There were 16 studies (2124 patients; three intervention nodes) regarding tube placement durations with well consideration of transitivity conception [8,9,11,12,13,15,17,18,19,21,22,23,24,25,26,27] (Figure 2B). Figure 3B shows the chest tube placement duration in which suction CTDS was used as a comparator. Digital CTDS significantly reduced the chest tube placement duration (MD: −0.68; 95% CI: −1.32–−0.04; P-score: 0.87), while the role of non-suction CTDS for chest tube placement duration remained inconclusive (Figure 3B and Figure 4). Appendix A shows the details of the head-to-head comparisons of outcomes.

### 3.4. Results of Prolonged Air Leak

There were 14 studies (2709 patients; three intervention nodes) regarding the occurrence of prolonged air leak with well consideration of transitivity conception [7,8,10,12,13,14,15,16,17,19,20,21,24,27] (Figure 2C). Figure 3C shows the results of prolonged air leak in which suction CTDS was used as a comparator. Although digital and non-suction CTDSs had a positive impact in preventing prolonged air leak, both did not reach a statistical significance (digital: OR, 0.76; 95% CI: 0.42–1.39; non-suction: OR, 0.95; 95% CI: 0.56–1.62). The P-score of digital CTDSs was 0.78, which was the highest among the interventions, followed by non-suction CTDSs (P-score, 0.41) (Figure 4). Appendix A shows the details of the head-to-head comparisons of outcomes.

### 3.5. Inconsistency, Risk of Bias, and Publication Bias

Net-split plots were used to present direct and indirect evidence in the NMA (Appendix A). The inconsistencies in these outcomes were evaluated using the design-by-treatment interaction (Appendix A) model and node-splitting model (Appendix A). In the examination of inconsistency, for both the length of hospital stay and chest tube placement duration, there was inconsistency when using the design-by-treatment interaction model (overall inconsistency, *p*-value < 0.0001 for both), but not when using the node-splitting model (random effects, *p*-value = 0.4326 and 0.4824, respectively). With a similar method, for the prolonged air leak, no inconsistency was found either in the design-by-treatment interaction model (overall inconsistency, *p*-value = 0.0614) or the node-splitting model (random effects, *p*-value = 0.9362). The RoB assessment for enrolled RCTs is illustrated and summarized in Appendix A. Moreover, Appendix A presents the comparison-adjusted funnel plots of all outcomes, and these disclosed no significant visual asymmetries. The Egger’s test revealed a statistical significance, indicating no evidence of publication bias.

### 3.6. Sensitivity Analysis

In the sensitivity analysis, we omitted high risk-of-bias studies as demonstrated in Appendix A. However, by nature of the designs of the enrolled studies, blinding and allocation concealment would be difficult; thus, these two were not accounted for when assessing risk of bias. In the sensitivity analyses, digital CTDS had the highest ranking, consistent across all measured outcomes. There was also no inconsistency among the length of hospital stay, chest tube placement duration, and prolonged air leak in the sensitivity analyses, regardless of using the design-by-treatment interaction (overall inconsistency, *p*-value = 0.3111, 0.6478, and 0.1577, respectively) or the node-splitting model (random effects, *p*-value = 0.1132, 0.2622, and 0.8724, respectively; Appendix A).

## 4. Discussion

This NMA aimed to compare the clinical efficacy of digital, suction, and non-suction CTDS in terms of their impact on hospital stay length, chest tube placement duration, and prolonged air leak after pulmonary resection. After comprehensively reviewing major databases, only randomized controlled trials were included to obtain updated evidence. Based on the 21 RCTs reviewed, digital CTDS was significantly associated with shorter length of hospital stay (the primary outcome) and chest tube placement duration compared to suction and non-suction CTDS. On the other hand, all three types of CTDS had no significant differences in preventing prolonged air leak.

Associated complications after pulmonary resections are not uncommon, such as prolonged air leak, bleeding, atelectasis, and pneumonia, which accounted for an incidence of around 6–23%, 0.1–0.3%, 1–20%, and 3–25%, respectively [4,35,52]. In this network meta-analysis, a total of 14 RCTs reported the incidence of adverse events after pulmonary resections with diverse outcomes (2% to 61.54%) [7,8,13,15,17,18,19,20,21,22,24,25,26,27]. Generally, most associated complications after pulmonary resections were self-limited; a well-drained pleural cavity with a physiologically negative-pressured environment could eliminate residual space and eventually promote the symphysis of injured visceral pleura via the chest tube with its connected CTDS [2,3,4,35,53].

In addition to a thorough preoperative evaluation and meticulous intraoperative manipulation, proper postoperative care is also essential to achieve a successful pulmonary resection. Early mobilization and adequate pain relief via thoracic epidural anesthesia are crucial during postoperative phase. Retaining a single chest tube for anatomic pulmonary resection, routine avoidance of applying an external suction, and early chest tube removal are highly recommended for chest tube management [3]. Despite its clinical significance, a chest tube with a connected CTDS could cause pain, impair pulmonary function, and hinder patients from doing physical activities regardless of the related surgical approaches [2,3]. Moreover, such discomfort/inconvenience due to a prolonged chest tube retention, which is a common clinical scenario after pulmonary resections, will delay patients’ recovery. Hence, an early chest tube removal is the ultimate goal for postoperative care to enhance the recovery after pulmonary resection and reduce hospital stay and related costs [3].

Since 2007, digital CTDSs became popular for diminishing inter-observer variability in decision-making during air leak assessment. They also have a precise intrapleural pressure detection and can maintain a steady, negative-pressured intrapleural environment via an electronic sensor and a digital console [2,3,5,7,8,9,10,18,19,20,21,22,23,25,26,27]. Moreover, digital CTDS has greater portability and has no concerns regarding water splitting (in contrast to using water seal chest bottles), which may facilitate patients’ physical activities. The guidelines of enhanced recovery after lung surgery also mentioned the use of digital CTDSs based on a low level of evidence [3,8,9]. Moreover, some authors tried to expand its application in the conservative management of primary spontaneous pneumothorax, with favorable results of shorter hospital stay and chest tube placement duration compared to non-suction CTDSs [54]. Nevertheless, one should not neglect the possibility of persistent air leak after removing chest tubes and subsequently reinserting a new one while using digital CTDSs. It is postulated that poor wound healing, delayed pleural symphysis, or compromised lung re-expansion due to underlying pulmonary pathologies (smoking or emphysema) might lead to persistent air leak after chest tube removal [53].

Both digital and non-suction CTDS (*p* = 0.90 and 0.60, respectively) were significantly associated with shorter hospital stay than suction CTDS (Figure 3A and Figure 4). Digital CTDS was associated with a reduction in hospital stay of 1.4 days (mean difference (MD): −1.40, 95% confidence interval (CI): −2.20 to −0.60), whereas non-suction CTDS was associated with a reduction in hospital stay by 1.05 days (MD: −1.05, 95% CI: −1.91 to −0.18) compared to suction CTDS. Digital CTDS was beneficial in minimizing the duration of chest tube placement after pulmonary resection (*p* = 0.87; Figure 3B and Figure 4). Digital and non-suction CTDSs were both associated with a reduction in chest tube duration of 0.68 days (MD: −0.68, 95% CI: −1.32 to −0.04) and 0.45 days (MD: −0.45, 95% CI: −1.11 to 0.20), respectively, compared to suction CTDS. Logically, earlier chest tube removal will lead to shorter hospital stay after pulmonary resection, which is the primary outcome of this NMA. The difference of our findings between hospital stay and chest tube placement duration could be attributed to different reasons. First, the enrolled studies had different facilities and surgeons with various care protocols and experience. Second, the enrolled studies were 20 years apart at most (Marshall et al., 2002 [11] and Takamochi et al., 2018 [27]) and thus could have been affected by new developments in anesthesiologic and surgical techniques. As for prolonged air leak after pulmonary resections, digital CTDS had a positive impact based on a pairwise comparison, but this was not statistically significant (*p* = 0.78; OR: 0.76; 95% CI: 0.42–1.39; Figure 3C and Figure 4). Non-suction CTDSs also had a lower OR (OR: 0.95; 95% CI: 0.56–1.62) in preventing prolonged air leak than suction CTDS. This might not have a positive impact for those with prolonged air leak after pulmonary resection. These findings were compatible with the guidelines for enhanced recovery after lung surgery published in 2019 [3]. Routine application of an external suction is also no longer recommended during the postoperative phase after pulmonary resection [3].

The first meta-analysis regarding the different choices of CTDSs after pulmonary resections was conducted by Coughlin et al. (2012), where seven RCTs were enrolled (from 2001 to 2007). No significant differences were found in terms of the duration of air leak, incidence of prolonged air leak, duration of chest tube placement, and length of hospital stay between suction and non-suction CTDSs [28]. In 2019, Zhou et al. meta-analyzed 10 RCTs (1601 patients enrolled) for the same issues. Although some studies focused on using digital CTDSs for pulmonary resections, the significance of non-suction CTDSs or external suction still remained inconclusive through their meta-analysis. Nevertheless, the necessity of applying an external suction is selectively justified based on a residual or increasing pneumothorax following pulmonary resections [32]. Recently, the role of digital CTDSs after pulmonary resections becomes dominant for its increasing use. Zhou et al. (2018) and Wang et al. (2019) also favored the clinical significance of digital CTDSs for patients undergoing pulmonary resections to shorten the length of hospital stays, chest tube placement duration, and air leak duration compared to suction CTDSs [30,31].

Through a frequentist network meta-analysis, these three similar CTDSs were compared indirectly, as no published study was available at the same time. Moreover, more precise effect estimates could be obtained by jointly assessing the direct and indirect comparisons [55,56,57]. Here, a pairwise comparison of three different CTDSs was performed simultaneously with more RCTs and case volumes enrolled (21 RCTs with 3399 patients), and only RCTs were enrolled to increase the statistical power. On the other hand, the PRISMA statement was strictly followed to improve the reporting of systematic reviews [33]. Nevertheless, there still exist some limitations within this network meta-analysis. First, heterogenicity in the management protocol for chest tubes (Appendix A), surgical types (open thoracotomy/VATS), and the criteria of reported postoperative adverse events might lead to inaccuracy during data sorting and interpretation [35]. Moreover, the majority of enrolled patients underwent thoracotomy (1479 patients recorded in total), and the readers should be cautious to interpret the results in our study. Second, although there were RCTs, no completely double-blinded study design was available for participants undergoing pulmonary resections; the patients/clinicians could realize the precise CTDS they were using. This will inevitably lead to a performance bias [58]. Third, no data of long-term follow-up were pursued for these enrolled RCTs, which might lead to an inaccurate reported incidence of adverse events. Fourth, not all RCTs enrolled in this study disclosed the associated adverse events after the pulmonary resections, and some reported adverse events were cardiopulmonary complications only [7,8,10,13,14,15,16,17,19,20,21,24,27]. Using a popular, standardized classification, such as the Clavien–Dindo classification, to precisely categorize these adverse events after pulmonary resections is necessary [59]. Lastly, the individualized intraoperative manipulations (including pleural tenting), surgeons’ experiences/preferences for pulmonary resections, and accompanied surgical instruments/linear staplers/reinforcement medical devices/sealants for pulmonary resections might be diverse from 1995 to 2016, which was a relatively long time frame. This might affect the perioperative outcomes after pulmonary resections and determine the hospital stay with related cost accordingly. Therefore, it is necessary to interpret the results of this network meta-analysis with caution.

## 5. Conclusions

Based on this network meta-analysis, digital CTDS is a more feasible strategy than suction and non-suction CTDSs for patients undergoing pulmonary resections. Digital CTDS is associated with 0.68 days shorter chest tube duration and 1.4 days shorter hospital stay than the suction CTDS.

## Figures and Tables

**Figure 1 jpm-12-00512-f001:**
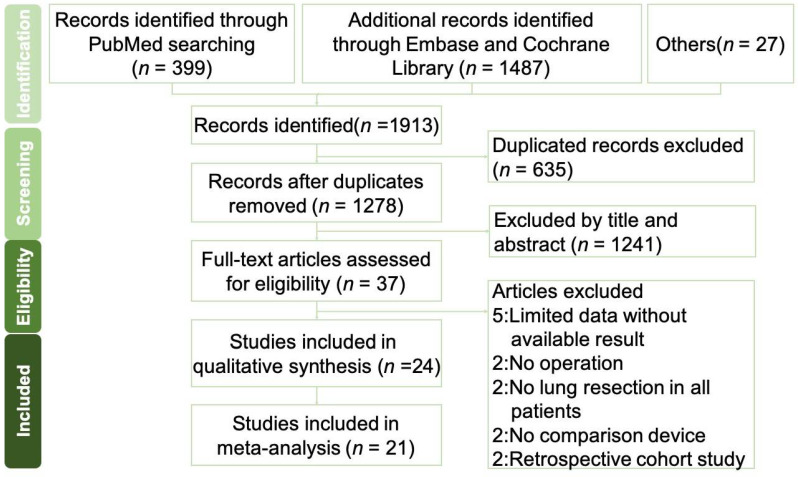
PRISMA flow diagram of the study selection. Flow diagram for the identification process of eligible studies.

**Figure 2 jpm-12-00512-f002:**
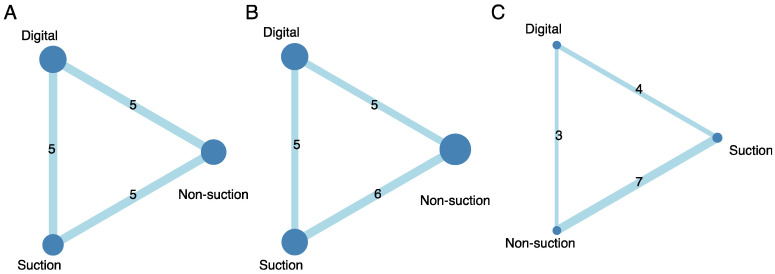
Network structure of an outcome measure. The size of the nodes represents the number of objectives involved in the treatment approach. The numbers on the lines represent the number of comparisons between each treatment approach. (**A**) The length of hospital stay; (**B**) chest tube placement duration; (**C**) prolonged air leak.

**Figure 3 jpm-12-00512-f003:**
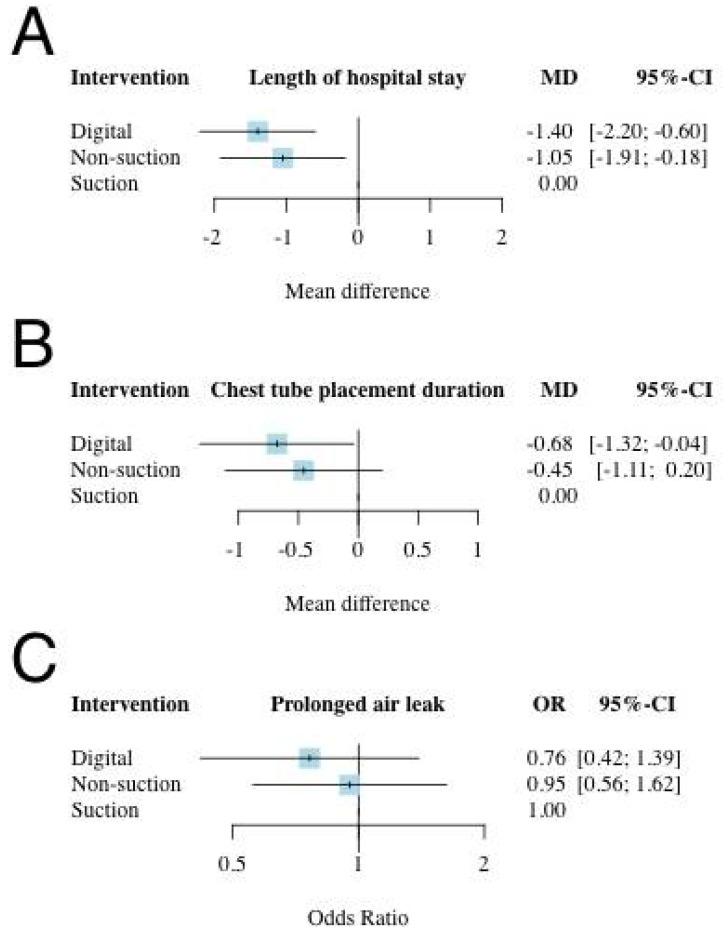
Network meta-analysis of (**A**) the length of hospital stay, (**B**) the chest tube placement duration, as well as (**C**) the prolonged air leak. The most beneficial intervention for the length of hospital stay was digital CTDS, which was 1.4 days shorter than the suction CTDSs (MD: −1.40; 95% CI: −2.20 to −0.60). Digital CTDS also significantly reduced chest tube placement duration by 0.68 days compared to suction CTDS (MD: −0.68; 95% CI: −1.32 to −0.04). Neither digital (OR: 0.76; 95% CI: 0.42–1.39) nor non-suction (OR: 0.95; 95% CI: 0.56–1.62) CTDS significantly reduced the risk of prolonged air leak. CI, confidence interval; MD, mean difference; OD, odds ratio.

**Figure 4 jpm-12-00512-f004:**
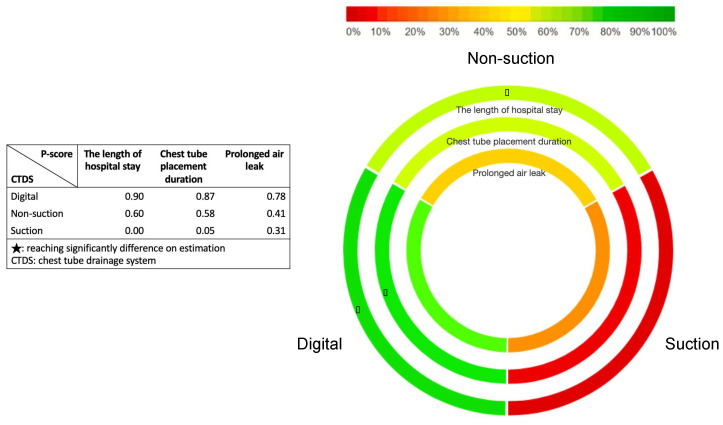
Rank-heat plot of P-score values among different chest tube drainage systems targeting the outcomes of length of hospital stay, chest tube placement duration, and prolonged air leak. Each slice of circle represents a different treatment. Treatments were ranked according to their P-score. A higher P-score (in green) denoted shorter hospital stay, shorter chest tube placement, and lower risk of prolonged air leak.

**Table 1 jpm-12-00512-t001:** Characteristics of enrolled studies.

Author, Year	Patient Number	Gender (Male/ Female%)	Age (Mean ±SD)	Comorbidities (Number)	Surgical Indication	Surgical Approach	Size of Drain	Resection Type	Reported Incidence of Adverse Events (%) and Associated Items
Marshall 2002 [11]	68	M: 49% F: 51%	63.4 ± 2.8	NR	Benign and malignant lung tumors	NR	NR	NR	NR
Ayed 2003 [12]	100	M: 94% F: 6%	23.0 ± 3.7	Patients with underlying lung disease were excluded.	Primary spontaneous pneumothorax	VATS: 100% Thoracotomy: 0%	28 Fr.	Wedge resection: 100%	NR
Brunelli 2004 [13]	145	M: 80.69% F: 19.31%	68.4 ± 9.2	NR	Nonsmall cell carcinoma.	VATS: 0% Thoracotomy: 100%	28 Fr.	Lobectomy or bilobectomy: 100%	24.83% (Atelectasis requiring bronchoscopy, pneumonia, pulmonary edema, adult respiratory distress syndrome, pulmonary embolism, pleural empyema, cardiac failure, arrhythmia requiring medical treatment, myocardial infarction, acute renal failure, and stroke.)
Alphonso 2005 [14]	254	M: 61.51% F: 38.49%	57.5 ± NR	Previous pneumothorax(71)	NR	VATS: 42.26% Thoracotomy: 57.74%	NR	Lobectomy: 46.44% Wedge resection: 44.77% Lung biopsy: 8.79%	NR
Brunelli 2005 [15]	94	M: 76.60% F: 23.40%	66.7 ± 10.1	NR	Nonsmall cell carcinoma.	VATS: 0% Thoracotomy: 100%	28 Fr.	Bilobectomy: 9.57% Lobectomy: 90.43%	24.47% (Atelectasis requiring bronchoscopy, pneumonia, pulmonary edema, adult respiratory distress syndrome, pulmonary embolism, pleural empyema, cardiac failure, arrhythmia requiring medical treatment, myocardial infarction, acute renal failure, and stroke)
Kakhki 2006 [16]	31	M: 70.97% F: 29.03%	36.8 ± 16.4	NR	NR	VATS: 0% Thoracotomy: 100%	NR	NR (excluding pneumonectomy or bronchoplasty)	NR
Cerfolio 2008 [7]	100	M: 51% F: 49%	62.0 ± NR	NR	Nonsmall cell carcinoma.	VATS: 0% Thoracotomy: 100%	NR	Lobectomy: 55% Segmentectomy: 16% Wedge resection: 29%	NR
Prokakis 2008 [17]	91	M: 63.74% F: 36.26%	59.5 ± 8.4	NR	Lung malignancies.	VATS: 0% Thoracotomy: 100%	32 Fr.	Bilobectomy: 14.29% Lobectomy: 85.71%	61.54% (Significant bleeding, sputum retention, atelectasis, pneumonia, cardiac arrhythmias, ventilatory support, empyema)
Brunelli 2010 [8]	166	M: 72.96% F: 27.04%	66.7 ± 10.9	Co-morbidity index(mean, (SD)): 1.69(1.65)	Lung cancer.	VATS: 0% Thoracotomy: 100%	28 Fr.	Lobectomy: 100%	15.06% (Only cardiopulmonary complications mentioned)
Filosso 2010 [9]	31	M: 67.74% F: 32.26%	69.6 ± 3.4	NR	Lung cancer.	VATS: 0% Thoracotomy: 100%	24 and 28 Fr.	Lobectomy: 100%	NR
Bertolaccini 2011 [18]	100	M: 59% F: 41%	65.5 ± 13.6	NR	NR	NR	24 and 28 Fr.	Lobectomy: 48% Segmentectomy: 6% Wedge resection: 46%	2% (Reoperation for bleeding, and one for exploratory thoracotomy)
Marjański 2013 [21]	64	M: 59.38% F: 40.62%	63.0 ± 21.5	Htpertension (25) Diabetes mellitus (7) Cardiovascular disease (6)	Lung cancer.	VATS: 51.56% Thoracotomy: 48.44%	28 Fr.	Lobectomy: 100%	37.50% (Atrial fibrillation, atelectasis requiring bronchial aspiration, prolonged air leak, redrainage, bronchial stump fistula, or pneumonia)
Brunelli 2013 [19]	100	M: 70% F: 30%	67.3 ± 10.6	Diabetes mellitus (13) Cardiovascular disease (14)	Lung cancer.	VATS: 0% Thoracotomy: 100%	28 Fr.	Lobectomy: 100%	13% (Only mentioning other cardiopulmonary complications)
Leo 2013 [20]	500	M: 64.40% F: 35.60%	63.5 ± NR	Chronic obstructive lung disease (114) Diabetes mellitus (77)	NR	NR	28 Fr.	NR	45.8% (Pneumothorax, subcutaneous emphysema, empyema without fistula, pulmonary pneumonia, atelectasis Requiring bronchoscopy, respiratory failure, atrial arrhythmia, pulmonary edema, cardiac ischemia, bronchial fistula, bleeding, reoperation for other reasons, laryngeal nerve palsy, and others)
Pompili 2014 [10]	390	M: 52.30% F: 47.70%	66.2 ± NR	NR	NR	VATS: 80.84% Thoracotomy: 19.16%	24 Fr.	Lobectomy: 85.30%% Segmentectomy: 14.70%	NR
Gilbert 2015 [22]	176	M: 36.36% F: 63.64%	68.0 ± NR	Co-morbidity index(mean):1	Benign or neoplastic lung disease	VATS: 72.09% Thoracotomy: 27.91%	NR	Lobectomy: 76.74% Segmentectomy: 23.26%	13.64% (New or worsening pneumothorax and/or increasing subcutaneous emphysema requiring chest tube reinsertion)
Lijkendijk 2015 [23]	105	M: 37.14% F: 62.86%	68.3 ± NR	NR	Lung cancer.	VATS: 39.04% Thoracotomy: 60.96%	24 Fr.	Lobectomy: 100%	NR
Gocyk 2016 [24]	254	M: 62.20% F: 37.80%	60.3 ± NR	NR	Malignant, benign and metastatic lung tumors.	NR	NR	Lobectomy: 55.51% Wedge resection: 44.49%	5.91% (Empyema, residual pneumothorax, peritonitis due to colon necrosis)
Chiappetta 2017 [25]	95	M: 51.58% F: 48.42%	63.6 ± 13.0	Htpertension (45) Diabetes mellitus (9) Cardiovascular disease (7) Chronic obstructive lung disease (26)	Benign or malignant lung disease	NR	28 Fr.	Lobectomy: 52.63% Wedge resection: 47.37%	2.11% (Reopening after clamping test, complication after chest tube removal)
Plourde 2018 [26]	215	M: 43.26% F: 56.74%	67.5 ± 9.3	NR	Benign or malignant lung tumors	VATS: 83.72% Thoracotomy: 16.28%	28 Fr.	Lobectomy: 93.49% Segmentectomy: 4.19% Wedge resection: 2.32%	5.12% (Pneumothorax, hemothorax, and empyema after tube removal)
Takamochi 2018 [27]	320	M: 50.31% F: 49.69%	67.3 ± 11.7	Diabetes mellitus (36) Cardiovascular disease (14) Cerebrovascular disease (7) Chronic obstructive lung disease (82) Interstitial pneumonia (28)	Malignant, benign and metastatic lung tumors.	VATS: 0% Thoracotomy: 100%	NR	Lobectomy: 79.26% Segmentectomy: 20.74%	21.25% (Pneumonia, atelectasis, bleeding, arrhythmia, chylothorax, and others)

**F** = female; **M** = male; **NR** = not recorded; **SD** = standard deviation; **VATS** = video-assisted thoracoscopic surgery; **Fr.** = French.

## Data Availability

Data is contained within the article or Appendix A.

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
