# Peer review of "Promising Effects of Digital Chest Tube Drainage System for Pulmonary Resection: A Systematic Review and Network Meta-Analysis"

_jpm, 2022, doi:10.3390/jpm12040512_

Round 1

Reviewer 1 Report

This manuscript presented by Po-Chih Chang et al. provides a very detailed and interesting synthesis of literature on performances and outcomes of different type of chest drain systems in surgical population 

The paper is so elegant and has been well written, but i would like to express some major concerns:

  1. It would be noted that most of data coming from studies, where thoracotomy represented a vast majority of interventions; thus, it would be suitable to consider this aspect for results interpretation, mentioning it into Discussion section.
  2. It may be noteworthy to provide data about comorbidities of patients and Indications for surgery (i.e: lung cancer, lung abscess etc..) 
  3. I would suggest adding, if available, details on size of drain used in included studies, being the latter a not negligible factor, influencing drainage power and performance of chest drain system regardless of suction

Author Response

Reviewer 1

This manuscript presented by Po-Chih Chang et al. provides a very detailed and interesting synthesis of literature on performances and outcomes of different type of chest drain systems in surgical population 

The paper is so elegant and has been well written, but i would like to express some major concerns:

1. It would be noted that most of data coming from studies, where thoracotomy represented a vast majority of interventions; thus, it would be suitable to consider this aspect for results interpretation, mentioning it into Discussion section.

Point to point response : 

Thank you for your valuable opinion. We have added the probable effect of thoracotomy in the limitation part of the discussion section. 

Revised in the manuscript: Discussion section, limitation. 

​​Moreover, the majority of enrolled patients underwent thoracotomy ( 1479 patients recorded in total), and the readers should be cautious to interpret the results in our study.

2. It may be noteworthy to provide data about comorbidities of patients and Indications for surgery (i.e: lung cancer, lung abscess etc..) 

Point to point response : 

We were very grateful for your suggestion. We have added the comorbidities of patients and indications for the surgery in Table 1. 

3. I would suggest adding, if available, details on size of drain used in included studies, being the latter a not negligible factor, influencing drainage power and performance of chest drain system regardless of suction

Point to point response : 

We appreciate your valuable suggestions very much. We have added the size of drains used in the included studies in Table 1. We also noted that most of the size of drain used in included studies was 28 Fr.. 

Reviewer 2 Report

Chang et al. here present a systematic review and MA on the type of optimal chest tube drainage system (CTDS; digital, non-suction, suction) after lung resections with regard to length of hospital stay (LOS), chest tube placement duration and presence of prolonged air leak.

21 RCTs with 3399 patients are analyzed for this purpose. The authors thus contribute to the generation of level 1 evidence in this area. As a result, digital CTDS appear to be beneficial in terms of shorter hospital stay and shorter duration of CTDS.

As far as I know, the work is the first to carry out a systematic comparison of 3 intervention groups and the question is therefore of high clinical relevance for the concrete everyday clinical practice of surgeons.

The question is clearly defined and the results are adequately presented and discussed. There are few questions that could further increase the quality and understanding of the article (minor revisions).

Methods: Please describe if digital CTDS is a suction or non-suction drainage system?

Results p 7, l 206/207: In „Methods“ it is stated that only RTCs with comparison of CTDS after lung resections were included. Why is therethen  a total of 3399 patients, of whom only 2326 had an operation? Please describe the difference.

Figure 4: What does the small square in the outermost three circles mean, should there be a * instead?

Results p 10, l 265: Please evaluate the use of the word "similarly", isn't there a contradiction to the previous sentence?

Discussion p 10, l 296/297: What does "self-limited" mean in this context?

Discussion p 11, l 337/338: Please explain which "difference" is meant here, the heterogeneity regarding LOS and chest tube duration between the various RCTs included?

Author Response

Reviewer 2

Chang et al. here present a systematic review and MA on the type of optimal chest tube drainage system (CTDS; digital, non-suction, suction) after lung resections with regard to length of hospital stay (LOS), chest tube placement duration and presence of prolonged air leak.

21 RCTs with 3399 patients are analyzed for this purpose. The authors thus contribute to the generation of level 1 evidence in this area. As a result, digital CTDS appear to be beneficial in terms of shorter hospital stay and shorter duration of CTDS.

As far as I know, the work is the first to carry out a systematic comparison of 3 intervention groups and the question is therefore of high clinical relevance for the concrete everyday clinical practice of surgeons.

The question is clearly defined and the results are adequately presented and discussed. There are few questions that could further increase the quality and understanding of the article (minor revisions).

1. Methods: Please describe if digital CTDS is a suction or non-suction drainage system?

Point to point response : 

Thank you very much for your valuable opinion. The digital CTDS a external suction system applied to the chest drains. In the method, study selection section, we also defined and explained three types of CTDS, which we also presented below:

Three different types of CTDS were identified and analyzed: digital, suction, and non-suction CTDS. Digital CTDS had digital sensors continuously monitoring pleural pressure and air flow, and applied an external suction accordingly, which would keep pleural pressure in a steady state. Suction CTDS had a fixed external suction force. Non-suction CTDS included those without an external suction force. The type of CTDS used in each study is described in Supplementary Table 3.

Revised in the manuscript: Method, study selection section

Three different types of CTDS were identified and analyzed: digital, suction, and non-suction CTDS. Digital CTDS had digital sensors continuously monitoring pleural pressure and air flow, and applied an external suction accordingly, which would keep pleural pressure in a steady state. Suction CTDS had a fixed external suction force. Non-suction CTDS included those without an external suction force. The type of CTDS used in each study is described in Supplementary Table 3.

2. Results p 7, l 206/207: In „Methods“ it is stated that only RTCs with comparison of CTDS after lung resections were included. Why is therethen  a total of 3399 patients, of whom only 2326 had an operation? Please describe the difference.

Point to point response : 

We appreciate your valuable comments very much. In the enrolled studies, there were a total of 3399 patients enrolled, but only 2326 patients underwent operation. The discrepancy occurred due to loss of data or drop out of the trials. For data extraction and analysis, an intention-to-treat principle was followed to extract the participant-related information in the enrolled RCTs

3. Figure 4: What does the small square in the outermost three circles mean, should there be a * instead?

Point to point response : 

We apologize for that sincerely, and we are very grateful for your opinion. the small square in Figure 4 should be presented as * instead. This mistake seemed to happen during the PDF formation, and the word manuscript disclosed * correctly. We are really sorry for that. 

4. Results p 10, l 265: Please evaluate the use of the word "similarly", isn't there a contradiction to the previous sentence?

Point to point response : 

We are really sorry for this unclear sentence here. We tried to express that we used a similar method to examine the inconsistency. Therefore, we rewrote our sentence as below: With a similar method, for the prolonged air leak, no inconsistency was found either in the design-by-treatment interaction model (overall inconsistency, p-value = 0.0614) or the node-splitting model (random effects, p-value = 0.9362).

Revised in the manuscript: Result, Inconsistency, risk of bias, and publication bias section

With a similar method, for the prolonged air leak, no inconsistency was found either in the design-by-treatment interaction model (overall inconsistency, p-value = 0.0614) or the node-splitting model (random effects, p-value = 0.9362).

5. Discussion p 10, l 296/297: What does "self-limited" mean in this context?

Point to point response : 

We are very grateful for your valuable comment. In this context, we tried to present that most associated complications after pulmonary resections were self-limited, and were able to be managed with a well-drained pleural cavity with a physiologically negative-pressured environment. We then also added “associated complications after pulmonary resection” after the “most” word.

Revised in the manuscript: Discussion

Generally, most associated complications after pulmonary resections were self-limited; a well-drained pleural cavity with a physiologically negative-pressured environment could eliminate residual space and eventually promote the symphysis of injured visceral pleura via the chest tube with its connected CTDS2-4,35,53.

6. Discussion p 11, l 337/338: Please explain which "difference" is meant here, the heterogeneity regarding LOS and chest tube duration between the various RCTs included?

Point to point response : 

Thank you for your valuable opinion. We are really sorry for this unclear sentence. The “difference” between hospital stay and chest tube placement duration could be at-tributed to different reasons. In this sentence, we tried to explain our findings in this network meta-analysis: Digital CTDS is associated with 0.68 days shorter chest tube duration and 1.4 days shorter hospital stay than the suction CTDS. Logically, earlier chest tube removal will lead to shorter hospital stay after pulmonary resection, but digital CTDS is associated with almost twice more shorter hospital stay than shorter chest tube duration. Therefore, we rewrote our sentence as below: The difference of our findings between hospital stay and chest tube placement duration could be at-tributed to different reasons.

Revised in the manuscript: Discussion

Logically, earlier chest tube removal will lead to shorter hospital stay after pulmonary resection, which is the primary outcome of this NMA. The difference of our findings between hospital stay and chest tube placement duration could be attributed to different reasons.

Round 2

Reviewer 1 Report

Much appreciated changes provided to main text, but some minor concerns remain:

- In table 1 Column "Surgical Indication" I would suggest a simpler form to present indications, removing the beginning sentence "Surgical intervention for...." and keeping just specific conditions that led patients to surgery 

- Page 11, line 403: Please change "CONCLUSSION" with  "CONCLUSION"

Author Response

Reviewer 1, Round 2

Much appreciated changes provided to main text, but some minor concerns remain:

Response:

Thank you very much for providing us this valuable opportunity to revise our manuscript.

- In table 1 Column "Surgical Indication" I would suggest a simpler form to present indications, removing the beginning sentence "Surgical intervention for...." and keeping just specific conditions that led patients to surgery

Response:

We appreciate your valuable recommendation. We have revised the table 1 accordingly. Thank you very much.

Revised in the manuscript, Table 1:

Author, year

Patient number

Gender (Male/Female %)

Age (mean ± SD)

Comorbidities(number)

Surgical Indication

Surgical approach

Size of drain

Resection type

Reported incidence of adverse events (%) and associated items

Marshall

2002

68

M:49%

F:51%

63.4 ± 2.8

NR

Benign and malignant lung tumors

NR

NR

NR

NR

Ayed

2003

100

M:94%

F:6%

23.0 ± 3.7

Patients with underlying lung disease were excluded.

Primary spontaneous pneumothorax

VATS: 100%

Thoracotomy: 0%

28 Fr.

Wedge resection: 100%

NR

Brunelli

2004

145

M:80.69%

F:19.31%

68.4 ± 9.2

NR

Nonsmall cell carcinoma.

VATS: 0%

Thoracotomy: 100%

28 Fr.

Lobectomy or bilobectomy: 100%

24.83%

(Atelectasis requiring bronchoscopy, pneumonia, pulmonary edema, adult respiratory distress syndrome, pulmonary embolism, pleural empyema, cardiac failure,

arrhythmia requiring medical treatment, myocardial infarction, acute renal failure, and stroke.)

Alphonso 2005

254

M:61.51%

F:38.49%

57.5± NR

Previous pneumothorax(71)

NR

VATS: 42.26%

Thoracotomy: 57.74%

NR

Lobectomy: 46.44%

Wedge resection: 44.77%

Lung biopsy: 8.79%

NR

Brunelli

2005

94

M:76.60%

F:23.40%

66.7 ± 10.1

NR

Nonsmall cell carcinoma.

VATS: 0%

Thoracotomy: 100%

28 Fr.

Bilobectomy: 9.57%

Lobectomy: 90.43%

24.47%

(Atelectasis requiring bronchoscopy, pneumonia, pulmonary edema, adult respiratory distress syndrome, pulmonary embolism, pleural empyema, cardiac failure, arrhythmia requiring medical treatment, myocardial infarction, acute renal failure, and stroke)

Kakhki

2006

31

M:70.97%

F:29.03%

36.8 ± 16.4

NR

NR

VATS: 0%

Thoracotomy: 100%

NR

NR (excluding pneumonectomy or bronchoplasty)

NR

Cerfolio

2008

100

M:51%

F:49%

62.0 ± NR

NR

Nonsmall cell carcinoma.

VATS: 0%

Thoracotomy: 100%

NR

Lobectomy: 55%

Segmentectomy: 16%

Wedge resection: 29%

NR

Prokakis

2008

91

M:63.74%

F:36.26%

59.5 ± 8.4

NR

Lung malignancies.

VATS: 0%

Thoracotomy: 100%

32 Fr.

Bilobectomy: 14.29%

Lobectomy: 85.71%

61.54%

(Significant bleeding, sputum retention, atelectasis, pneumonia, cardiac arrhythmias, ventilatory support, empyema)

Brunelli

2010

166

M:72.96%

F:27.04%

66.7 ± 10.9

Co-morbidity index(mean, (SD)): 1.69(1.65)

Lung cancer.

VATS: 0%

Thoracotomy: 100%

28 Fr.

Lobectomy: 100%

15.06%

(Only cardiopulmonary complications mentioned)

Filosso

2010

31

M:67.74%

F:32.26%

69.6 ± 3.4

NR

Lung cancer.

VATS: 0%

Thoracotomy: 100%

24 and 28 Fr.

Lobectomy: 100%

NR

Bertolaccini 2011

100

M:59%

F:41%

65.5 ± 13.6

NR

NR

NR

24 and 28 Fr.

Lobectomy: 48%

Segmentectomy: 6%

Wedge resection: 46%

2%

(Reoperation for bleeding, and one for exploratory thoracotomy)

Marjański 2013

64

M:59.38%

F:40.62%

63.0 ± 21.5

Htpertension(25)

Diabetes mellitus(7)

Cardiovascular disease(6)

Lung cancer.

VATS: 51.56%

Thoracotomy: 48.44%

28 Fr.

Lobectomy: 100%

37.50%

(Atrial fibrillation, atelectasis requiring bronchial aspiration, prolonged air leak, redrainage, bronchial stump fistula, or pneumonia)

Brunelli

2013

100

M:70%

F:30%

67.3 ± 10.6

Diabetes mellitus(13)

Cardiovascular disease(14)

Lung cancer.

VATS: 0%

Thoracotomy: 100%

28 Fr.

Lobectomy: 100%

13%

(Only mentioning other cardiopulmonary complications)

Leo

2013

500

M:64.40%

F:35.60%

63.5 ± NR

Chronic obstructive lung disease(114)

Diabetes mellitus(77)

NR

NR

28 Fr.

NR

45.8%

(Pneumothorax, subcutaneous emphysema, empyema without fistula, pulmonary pneumonia, atelectasis

Requiring bronchoscopy, respiratory failure, atrial arrhythmia, pulmonary edema, cardiac ischemia, bronchial fistula, bleeding, reoperation for other reasons, laryngeal nerve palsy, and others)

Pompili

2014

390

M:52.30%

F:47.70%

66.2 ± NR

NR

NR

VATS: 80.84%

Thoracotomy: 19.16%

24 Fr.

Lobectomy: 85.30%%

Segmentectomy: 14.70%

NR

Gilbert

2015

176

M:36.36%

F:63.64%

68.0 ± NR

Co-morbidity index(mean):1

Benign or neoplastic lung disease

VATS: 72.09%

Thoracotomy: 27.91%

NR

Lobectomy: 76.74%

Segmentectomy: 23.26%

13.64%

(New or worsening pneumothorax and/or increasing subcutaneous emphysema requiring chest tube reinsertion)

Lijkendijk 2015

105

M:37.14%

F:62.86%

68.3 ± NR

NR

Lung cancer.

VATS: 39.04%

Thoracotomy: 60.96%

24 Fr.

Lobectomy: 100%

NR

Gocyk

2016

254

M:62.20%

F:37.80%

60.3 ± NR

NR

Malignant, benign and metastatic lung tumors.

NR

NR

Lobectomy: 55.51%

Wedge resection: 44.49%

5.91%

(Empyema, residual pneumothorax, peritonitis due to colon necrosis)

Chiappetta 2017

95

M:51.58%

F:48.42%

63.6 ± 13.0

Htpertension(45)

Diabetes mellitus(9)

Cardiovascular disease(7)

Chronic obstructive lung disease(26)

Benign or malignant lung disease

NR

28 Fr.

Lobectomy: 52.63%

Wedge resection: 47.37%

2.11%

(Reopening after clamping test, complication after chest tube removal)

Plourde

2018

215

M:43.26%

F:56.74%

67.5 ± 9.3

NR

Benign or malignant lung tumors

VATS: 83.72%

Thoracotomy: 16.28%

28 Fr.

Lobectomy: 93.49%

Segmentectomy: 4.19%

Wedge resection: 2.32%

5.12%

(Pneumothorax, hemothorax, and empyema after tube removal)

Takamochi 2018

320

M:50.31%

F:49.69%

67.3 ± 11.7

Diabetes mellitus(36)

Cardiovascular disease(14)

Cerebrovascular disease(7)

Chronic obstructive lung disease(82)

Interstitial pneumonia(28)

Malignant, benign and metastatic lung tumors.

VATS: 0%

Thoracotomy: 100%

NR

Lobectomy: 79.26%

Segmentectomy: 20.74%

21.25%

(Pneumonia, atelectasis, bleeding, arrhythmia, chylothorax, and others)

- Page 11, line 403: Please change "CONCLUSSION" with  "CONCLUSION"

Response:

We apologized for the wrong spelling of Conclusion. We have corrected it. Thank you very much again. 

Revised in the manuscript, Conclusion section.

CONCLUSION

Based on this network meta-analysis, digital CTDS is a more feasible strategy than suction and non-suction CTDSs for patients undergoing pulmonary resections. Digital CTDS is associated with 0.68 days shorter chest tube duration and 1.4 days shorter hospital stay than the suction CTDS.
